# Mandatory membership of community-based mutual health insurance in Senegal: A national survey

**Valéry Ridde** [1,2]*, **Ibrahima Gaye**[2], **Bruno Ventelou**[3], **Elisabeth Paul**[4], **Adama Faye**[2]

**1** CEPED, IRD-Université de Paris, ERL INSERM SAGESUD, Paris, France, **2** Institute of Health and Development (ISED), Cheikh Anta Diop University, Dakar, Senegal, **3** French National Center for Scientific Research (CNRS), Aix-Marseille School of Economics, Aix Marseille University, Marseille, France, **4** Université libre de Bruxelles, School of Public Health, Brussels, Belgium

* valery.ridde@ird.fr

**Data Availability Statement:** All data are presented in the paper.

**Funding:** This research is part of the Unissahel program (Universal Health Coverage in Sahel:

## Abstract

With the low adherence to voluntary mutual health insurance, Senegal's policymakers have sought to understand the feasibility of compulsory health insurance membership. This study aims to measure the acceptability of mandatory membership in community-based mutual health insurance (CBHI) and to understand its possible administrative modalities. The study consists of a national survey among a representative population sample selected by marginal quotas. The survey was conducted in 2022 over the phone, with a random composition method involving 914 people. The questionnaire measured the socio-economic characteristics of households, their level of acceptability concerning voluntary and compulsory membership, and their level of confidence in CBHIs and the health system. Respondents preferred voluntary (86%) over mandatory (70%) membership of a CBHI. The gap between voluntary and compulsory membership scores was smaller among women (p = 0.040), people under 35 (p = 0.033), and people with no health coverage (p = 0.011). Voluntary or compulsory membership was correlated (p = 0.000) to trust in current CBHIs and health systems. Lack of trust in the CBHI management has been more disadvantageous for acceptance of the mandatory than the voluntary membership. No particular preference emerged as the preferred administrative channel (e.g. death certificate, identity card, etc.) to enforce the mandatory option. The results confirmed the well-known challenges of building universal health coverage based on CBHIs—a poorly appreciated model whose low performance reduces the acceptability of populations to adhere to it, whether voluntary or mandatory. Suppose Senegal persists in its health insurance approach. In that case, it will be essential to strengthen the performance and funding of CBHIs, and to gain population trust to enable a mandatory or more systemic membership.

## Introduction

In their search for solutions to provide their citizens with health insurance, African countries face many challenges and have tested various strategies [1]. In Francophone West Africa,

https://www.unissahel.org) funded by the Agence Française de Développement (AFD) and another grant has been supported by the Wallonie Bruxelles International (WBI). The opinions expressed are exclusive of the authors and do not reflect the official position of the AFD and WBI. The funders had no role in study design, data collection and analysis, decision to publish, or preparation of the manuscript.

**Competing interests:** The authors have declared that no competing interests exist.

community-based health insurance (CBHI) has long been the preferred instrument for including populations in the informal and rural sectors [2–4]. While evidence shows CBHIs protect their members [3, 5], their penetration rates are very low. In Senegal, less than 5% of the population are affiliated with a CBHI [6, 7] despite subsidized membership and the fact that they have been at the heart of the health financing policy for over a decade [8–10]. As international organizations and many experts explain, the voluntary nature of membership and their amateurish management cause people to question the usefulness of CBHIs as a tool for progressing toward universal health coverage (UHC) [11–13]. Since 1952, the International Labour Organization (ILO) Convention 102 on Social Security has affirmed the importance of compulsory or automatic membership. However, the Regulation on Social Mutuality of the West African Economic and Monetary Union (WAEMU) recognizes the right to freely adhere to and choose a mutual organization.

Since 2014, Senegal has experimented with a different model of CBHI, consisting of departmental mutual health insurance managed by professionals, whose coverage rate demonstrates its relevance [14]. Despite being based on voluntary membership, they pave the way for expanding the system by adopting a mandatory membership. However, no research has yet been undertaken in Senegal to understand the social acceptability of such compulsory membership. The 2017 national health financing strategy and the 2019 National Health Development Programme (PNDSS 2019–2028) proposed to study the feasibility of such a mandatory approach [15, 16]. In particular, the PNDSS calls *for the promotion of 'mechanisms linking compulsory membership in an insurance scheme of one's choice and obtaining certain administrative documents and advantages (driving licence, etc.).'*[16]. In 2020, the President of the Republic suggested organizing a '*smart obligation*' by requesting proof of one's affiliation to an insurance scheme to benefit from micro-credit or to carry out administrative acts [17]. In Rwanda, for example, health insurance membership is compulsory, and many people are affiliated with a CBHI, although they are now more State than Community instruments [18]. Senegal was the first African country to ratify ILO Convention 102 in October 1962, but not about health insurance [19]. A national evaluation undertaken in 2021 [20] pressed Senegal to significantly reform its UHC strategies, particularly by reevaluating the systematic nature of membership in departmental mutual health insurance organizations.

With a descriptive and analytical cross-sectional study, this article aims to measure the propensity of the Senegalese population to accept a potential obligation to adhere to mutual health insurance and to understand the feasibility of administrative arrangements.

## Materials and methods

### Setting

The study was conducted in the 14 administrative regions of Senegal. The average age in Senegal is 19 years. Men account for 49.7% of the population. According to the *Autorité de Régulation des Télécommunications et des Postes du Sénégal*, the mobile phone usage rate is 119%. In Senegal, 4.5% of the household population is affiliated with a CBHI, 1.3% with professional mutual insurance, and 0.3% with private insurance in 2019 [7].

### Study design and population

This report is a descriptive and analytical cross-sectional study (see S1 Checklist). To our knowledge, this is the first study on the subject in Senegal and West Africa. Thus, we consider this study exploratory, seeking to answer our objective without formulating any hypotheses at this stage. The results of the study may be useful for subsequent hypothetico-deductive studies. The data were collected from April 26 to July 22, 2022. The study population comprised people

aged 18 and over in the general population with a mobile phone number and not already a member of a CBHI.

## Sampling

The study used a marginal quota sampling strategy [21]. To ensure a representative population sample, we performed a precision stratification by region, sex, and age group. Following an approach already used by our team for surveys during the pandemic [22, 23], we randomly generated a list of nine-digit phone numbers from the mobile phone numbers assignable in Senegal using the Random Dialing (RDD) method. We integrated this list into a Reactive Auto Dialer (RAD) to trigger calls automatically and optimally. A total of 914 people participated in the study.

## Data collection

Five research assistants speaking the six main languages of Senegal (French, Diola, Wolof, Serère, Pulaar, Soninké) collected the quantitative data using a close-ended structured questionnaire by telephone. They administered the questionnaire using tablets with the Open Data Kit (ODK) software. The final version of the questionnaire was tested during the assistants' training and then validated by the team members after several corrections. In addition to socio-economic variables (age, gender, education, marital status, assets), we conceptualized the collected variables according to the seven dimensions of the Sekhon et al. acceptability model (Table 1) [24], whose previous relevance was verified in the context of Senegal [22]. Each of the seven dimensions was appraised using a Likert scale ranging from one (do not agree at all) to five (strongly agree). These dimensions made it possible to assess the population's interest in joining a CBHI voluntarily and to determine perceptions of a possible

**Table 1. Operationalization of the seven dimensions of acceptability.**

| Dimension | Voluntary membership | Compulsory membership |
|---|---|---|
| Importance | I have understood the importance of joining mutual health insurance offered to all on a voluntary basis. | I have understood the importance of making membership in mutual health insurance compulsory in order to carry out administrative acts. |
| Effort | I will make a great deal of effort to be able to join mutual health insurance offered to all on a voluntary basis. | I will make a great deal of effort to comply with the obligation to join mutual health insurance to carry out administrative acts. |
| Perceptions | My feelings about the voluntary membership of mutual health insurance offered to all are positive. | My feelings about this obligation to join mutual health insurance to carry out administrative acts are positive. |
| Effectiveness | I believe that voluntary membership of mutual health insurance proposed to all will improve the health coverage of Senegalese people. | I believe that making membership of mutual health insurance compulsory to carry out administrative acts will improve the health coverage of Senegalese people. |
| Benefits | I think the benefits from voluntary membership in mutual health insurance offered to all are worth the investment I will have to put forth in order to get it. | I think that the benefits from making membership in mutual health insurance compulsory to carry out administrative acts are worth the investment I will have to put forth in order to respect it. |
| Trust | I am confident in my ability to join mutual health insurance offered to all on a voluntary basis. | I am confident in my ability to respect the obligation to join mutual health insurance to carry out administrative acts. |
| Personal values | Joining mutual health insurance offered to all on a voluntary basis aligns with my values. | Making membership in mutual health insurance compulsory to carry out administrative acts aligns with my values. |

---

**Box 1. Definition of a CBHI used in the questionnaire.**

Mutual health insurance is an organization that receives voluntary (thus non-compulsory) contributions from its members in advance. People pay a membership fee once in their lifetime at the beginning; then, they pay a membership fee each year. It covers a large part (often 80%) of healthcare costs for sick members and their dependents. It does not reimburse contributions if the member or her/his dependents have not been sick.

---

mandatory adherence (Box 1) conditioning the access to a number of administrative acts (compulsory membership). The proposed channel list corresponds to Senegal's most frequent administrative actions. The role of trust in the acceptability of measures has been revealed in our previous studies in Senegal [22] as well as the literature on CBHI membership [3] and health systems [25]. Thus, we added two questions about trust in i) existing CBHIs, and ii) the health system and its ability to protect patients against catastrophic expenditure in the UHC context.

## Data analysis

We carried out a descriptive study of the different variables. Categorial variables were described through frequency measurements with their confidence intervals, with a 95% confidence level, and quantitative data through central trend and dispersion measurements. A cross-sectional analysis was performed between the dependent and relevant independent variables. Appropriate statistical tests (Student, Mann-Whitney) were used to measure associations with an alpha risk of 5%. Correlations were computed to study the links between the different dimensions. The well-being score was calculated in three stages: I) all household goods for which information has been collected were assigned a weight generated after principal component analysis; ii) the scores were summed per household and standardized with respect to a standard normal distribution with a mean of 0 and a standard deviation of 1; and iii) households were ranked in ascending order and the sample was divided into quintiles, i.e. five groups with the same number of people in each. The 20% of the population with the lowest total wealth scores became the individuals in the lowest quintile; the next 20% were considered the second wealth quintile, and so on.

The acceptability of membership in CBHIs was evaluated by constructing two scores: one for voluntary membership and one for mandatory membership. These scores were obtained by combining individuals' degrees of approval for membership of a CBHI offered to all voluntarily, with the benefits derived from such membership and the compatibility between such membership and personal values. The scores thus obtained, whose consistency was verified using the Cronbach Alpha Index (0.989 for compulsory membership and 0.939 for voluntary membership), were standardized to vary between 0 and 100. A Tobit regression was carried out to analyze the relationship between confidence and membership, controlling for possible confounding bias. All analyses were carried out with the software STATA 14.

## Ethics approval, consent to participate, competing Interests

All individuals were informed of the ethical issues and were allowed to withdraw from the study at any time. They all agreed to participate through verbal informed consent recorded by the interviewer under the supervision of the researchers. No minor participated in this

research. The research was accepted by Senegal's National Health Research Ethics Committee (42/MSAS/CNERS/SP/2021). The authors have declared that no competing interests exist. See S1 Questionnaire for inclusivity for global health research process.

## Results

Most of the sample comprised women (51.3%) and young people under 35 (54.8%). A large proportion (44.7%) were uneducated (Table 2).

Respondents were more in favour of voluntary membership (86%) of a CBHI than of compulsory membership (70%). The rejection of mandatory membership was significantly more predominant among men (59%), those with higher education (68%), those already utilizing non-mutual health insurance (75%), those distrusting the current functioning of CBHI (76%), and those with low confidence in the government's health system (57%).

The gap between voluntary and compulsory membership scores was significantly smaller among women (p = 0.040), people under 35 years (p = 0.033), and people with no health insurance (p = 0.011). The difference did not appear to be influenced by marital status (p = 0.096), education (p = 0.252), and income (p = 0.034).

Regarding both voluntary or compulsory membership, a sense of trust played an important role. The correlation between trust and membership was unambiguous (p = 0.000). The more people were confident in the current functioning of CBHI (Fig 1) or in the way in which the State takes care of its health system (so that it is accessible to all, of quality, and without

**Table 2. Characteristics of respondents (n = 914).**

| Variable | Number | Percentage |
|---|---|---|
| **Gender** | | |
| Female | 469 | 51.3% |
| Male | 445 | 48.7% |
| **Age category** | | |
| Under 35 years | 501 | 54.8% |
| 35 years and over | 413 | 45.2% |
| **Level of education** | | |
| Without education | 409 | 44.7% |
| Primary | 163 | 17.8% |
| Secondary | 239 | 26.1% |
| Tertiary | 103 | 11.3% |
| **Marital status** | | |
| Married monogamous | 451 | 49.3% |
| Married polygamous | 142 | 15.5% |
| Divorced | 22 | 2.4% |
| Widow(er) | 38 | 4.2% |
| Single | 258 | 28.2% |
| Refused to say | 3 | 0.3% |
| **Quintile of well-being** | | |
| Poorest | 182 | 20.0% |
| Poor | 183 | 20.0% |
| Average | 183 | 20.0% |
| Rich | 183 | 20.0% |
| Richest | 183 | 20.0% |
| **Total** | **914** | **100.0%** |

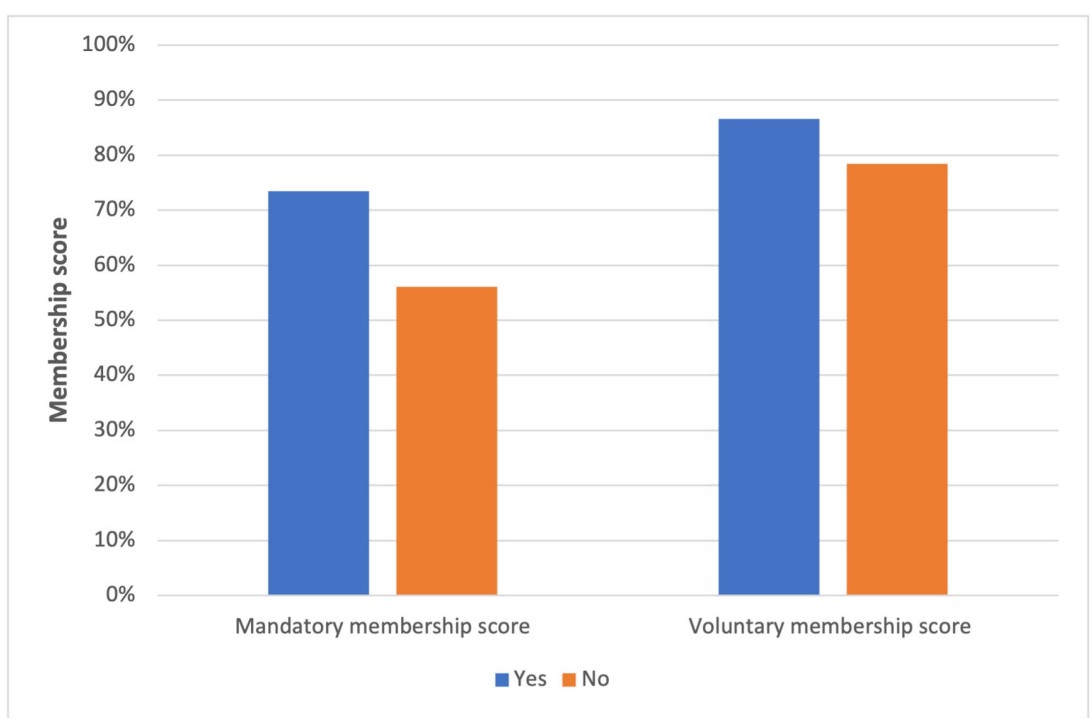

**Fig 1. Level of membership score according to confidence (yes/no) in the government as for the management of CBHI.**
Note: Fig 1 shows that there are more respondents in favour of voluntary than mandatory membership.

financial consequences for service users) (Fig 2), the more they were in favour of (voluntary or compulsory) membership.

Table 3, with the comparison of columns 2 and 3, highlights the strong negative impact of the level of instruction on acceptability in case of a mandatory. Meanwhile, it confirms that pre-existing trust in CBHI management and the government health system management were factors of the acceptability score, all else being equal; there is no confusion effect with, e.g. the individuals' instruction level. Table 3 also reveals that the lack of trust in the CBHI management has been more deleterious for acceptability in the mandatory case (comparison of the mean of the two estimated slopes: -18.4 vs -9.5; although confidence intervals tend to overlap: -18.4***[IC: -28.75; -8.07]; -9.5***[IC: -14.66; -4.36]). In other words, should the government opt for a mandatory enrolment strategy, the prior construction of a strong population confidence in the system becomes crucial to obtain population adherence.

The study did not reveal any particular preference for the administrative channel used to enforce mandatory adherence. The level of acceptability (63% to 65%) was overall the same if payment would be requested when drawing up a death certificate, identity card, passport, birth certificate, or registration of children in school. Moreover, among those supporting compulsory membership, the higher their education degree, the less they accept this method—regardless of the proposed administrative channel (Table 4). There was no statistically significant difference in the economic level of the respondents.

## Discussion

This research is the first on the subject in Senegal. The data show that seven out of ten people accept the compulsory nature of membership in a CBHI. However, the more educated people are, the less they accept the compulsory nature. There was no preference for the administrative

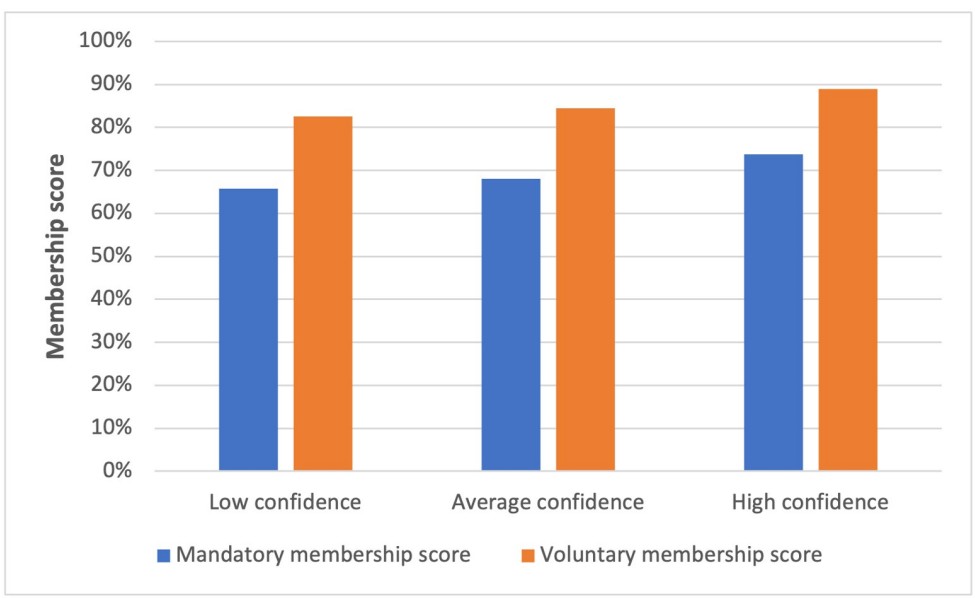

**Fig 2. Level of membership score according to confidence (low, medium, high) in the current health system.** Note: Fig 2 shows that the more confidence respondents have in the health system, the more likely they are to join the insurance.

channel (death certificate, identity card, etc.) for collecting this contribution. Finally, acceptance of the compulsory membership was very much associated with confidence in the current CBHI and health system management.

This study shows that many people declared *intending to* accept the voluntary nature of membership in CBHI. This high acceptance of the voluntary nature of the contribution has already been revealed in research on UHC in Burkina Faso [26]. Beyond the methodological challenges of this survey, the low *current* rate of the total membership in Senegal [1, 10] confirms the low attractivity of CBHIs to date. Beyond the technical and financial challenges, social representation vis-à-vis CBHIs in Africa, particularly in Senegal [27], influences their attractiveness, which does not seem sufficiently studied by researchers yet, nor considered by operators [28]. Overall, Senegal's CBHIs do not yet meet the populations' needs, they are still hardly known to them, and their functioning is not optimal [8, 10, 27, 29, 30]. Since the acceptability is even lower for the mandatory character of membership and closely linked to trust in current (health and CBHI) systems, strengthening CBHIs is undoubtedly a prerequisite before any reform.

The COVID-19 pandemic and State response measures, notably in Senegal, emphasized the importance of citizens' trust in the acceptability of public policies. As with this study, the more educated people are, the more they question the merits and ability of these State-organized measures [22]. The issues of solidarity and risk-sharing among population groups are at the heart of UHC debates, as surveys elsewhere in Africa show [26, 31]. It will be recalled that in Mali, when the government imposed compulsory insurance for academic professors, they opposed it, and the government had to step back [32]. In addition, considerable research on health systems also confirms the importance of trust between health professionals [25, 33] and also users for the effectiveness of health services as well as members of CBHIs [34], particularly in Senegal [27]. Research in Burkina Faso has shown how central this trust is to adherence to UHC financial principles [26]. This study confirms these results for Senegal.

**Table 3. Relationship between confidence level and membership score.**

| | Voluntary membership score | | Mandatory membership score | |
|---|---|---|---|---|
| **Confidence in CBHI management (Reference = Yes)** | | | | |
| No | -9,51*** [-14,66; -4,36] | | -18,41*** [-28,75; -8,07] | |
| | (2,62) | | (5,27) | |
| No opinion | -0,51 [-2,32;1,31] | | -7,38*** [-11,49; -3,28] | |
| | (0,92) | | (2,09) | |
| **Confidence in the government's healthcare system (Reference = Low confidence)** | | | | |
| Average confidence | | 2,40* [-0,04;4,85] | | 2,04 [-3,36;7,45] |
| | | (1,25) | | (2,75) |
| High confidence | | 7,47*** [5,05;9,88] | | 9,21*** [3,71;14,71] |
| | | (1,23) | | (2,80) |
| Age | -0,01 [-0,08;0,06] | -0,01 [-0,08;0,06] | -0,18** [-0,32;-0,03] | -0,16** [-0,30;-0,02] |
| | (0,04) | (0,04) | (0,08) | (0,07) |
| Sexe (Men) | -1,09 [-2,82;0,63] | -0,42 [-2,03;1,19] | -5,17*** [-9,03;-1,31] | -3,46* [-7,34;0,41] |
| | (0,88) | (0,82) | (1,97) | (1,97) |
| Level of education | 0,50 [-0,28;1,29] | 0,18 [-0,57;0,93] | -2,63*** [-4,51;0,75] | -2,69*** [-4,54;-0,84] |
| | (0,40) | (0,38) | (0,96) | (0,94) |
| **Financial situation (Reference = Very comfortable)** | | | | |
| Comfortable | 0,47 [-4,49;5,44] | 1,16 [-3,52;5,84] | -6,12 [-16,33;4,08] | -5,18 [-15,31;4,94] |
| | (2,53) | (2,39) | (5,20) | (5,16) |
| Poor | 3,49 [-1,58;8,56] | 3,80 [-1,03;8,64] | -1,39 [-12,14;9,36] | -0,75 [-11,45;9,95] |
| | (2,59) | (2,46) | (5,48) | (5,45) |
| Very Poor | 4,64 [-1,45;10,72] | 3,92 [-1,90;9,74] | -8,64 [-23,27;5,99] | -8,95 [-23,49;5,58] |
| | (3,10) | (2,96) | (7,45) | (7,45) |
| Constant | 86,41*** [81,06;91,77] | 81,31*** [75,67;86,95] | 97,06*** [83,64;110,47] | 85,16*** [71,73;98,59] |
| | (2,73) | (2,87) | (6,84) | (6,84) |

Robust standard errors in brackets

*** p<0,01,

** p<0,05,

*p<0,10

**Table 4. Preferred administrative channel for enforcing systematic membership for those in favour of the strategy.**

| | Not educated | Primary | Secondary | Tertiary | P value |
|---|---|---|---|---|---|
| Requesting a death certificate | 93.50% | 93.50% | 83.30% | 82.90% | 0.008 |
| Applying for a national identity card | 94.60% | 95.90% | 82.90% | 67.90% | 0.000 |
| Opening a micro-business | 92.40% | 97.50% | 86.70% | 75.90% | 0.000 |
| Applying for or renewing a passport | 92.30% | 94.70% | 84.60% | 75.10% | 0.013 |
| Applying for or renewing a driving licence | 92.40% | 95.80% | 87.20% | 67.00% | 0.000 |
| Paying income tax | 92.40% | 94.90% | 87.00% | 78.80% | 0.018 |
| Enrolling children in school | 94.60% | 96.90% | 90.20% | 87.80% | 0.199 |
| Requesting a birth certificate | 95.60% | 96.50% | 84.30% | 69.60% | 0.000 |

Expanding population health insurance coverage would improve efficiency and equity in a fragmented and underfunded Senegalese health system [35], where governance is questioned [9, 36]. This also requires improving the overall functioning of the health financing system and addressing the critical challenges in terms of fiscal space (where can resources for CBHI membership be found?), the strategic purchasing of health services and the operational efficiency of financial management [37]. In addition, evidence increasingly demonstrates the better performance of government tax-funded systems over those funded through contributory social insurance [38], which Senegal should consider. But the failure of CBHIs in Tanzania drove researchers to assert that '*making health insurance mandatory requires an adequate legal and regulatory framework to help safeguard the mandatory membership and protect a standard Minimum Benefits Package (MBP) and entitlement of members*' [39].

Following a recent evaluation [20], the Senegalese National Agency for Universal Health Coverage launched a significant reform in early 2023. First, it confirmed its announcements at the end of 2022, and explained that all communal CBHIs would be dissolved and that only departmental CBHIs would be organized nationwide. In addition to strengthening their professionalization, this levelling up in risk pooling has the potential to improve the financial sustainability of health insurance, as has long been proposed for the region [1, 40]. Furthermore, the National Agency announced the end of membership State subsidies. Indeed, at the outset of the expansion of the CBHI system, the State was aware of the challenges of affordability and the financial barrier that the premium could pose to increasing insurance coverage. Thus, the state subsidized up to 50% of the annual fees for contributors and up to 100% for non-contributors (worst-off and people with disabilities). It announced its intent to transform membership subsidy into purchasing CBHI members' healthcare consumption directly to health facilities. These two major changes have caused significant tensions with the National Union of CBHI. They may be the consequences of the fiscal space challenges or a change in government priorities. The future will tell us whether these reforms increase citizens' confidence in their health and CBHI systems and contribute to rendering membership more systematic. Even if the policy instrument is unchanged (insurance), it may illustrate a return of the State [41] in the field of social protection, as has long been demanded [42]. As with the success of CBHIs in Ethiopia, it is essential to give a '*signal of strong political and government commitment*' [43]. It may be time to invalidate the conclusion that '*health and health goals might be unimportant to a country's policy choices*' [44].

The study makes it possible to propose some involvement pathways for Senegal's public health system that UHC stakeholders discussed during a workshop organized by the National Agency in December 2022. First, it appears urgent to strengthen the performance of CBHIs and the insurance system in general before considering making it mandatory. The current reforms (professionalization, departmentalization, digitalization) should allow this improvement since the new governance structure gives room to community approaches and citizens, strengthening the collective identity that seems to have been lacking so far in Senegal [45]. Even if CBHIs disappear and are transformed into a health insurance fund, the question of subscription and financing will remain a central element: '*there is now general agreement in the literature that, for countries to make progress towards UHC, health financing needs to include aspects of both compulsion and redistribution*'[46]. Secondly, there is an urgent need to improve the public's confidence in CBHIs (including services rendered and reimbursements) and in the health and social protection systems. Finally, Senegal imposed declaring blood type on the new digital driving licence in 2019. Suppose all administrative channels seem possible to enforce the compulsory contribution for health insurance. In that case, it becomes urgent to check its technical feasibility and to manage specific communication processes adapted to the targets.

This research has several limitations. First, given the small percentage of people with health insurance in Senegal (exposure), the questions asked about contributions may have been misunderstood [26] even if we have clearly explained the definition of CBHIs (Box 1). In addition, the study focuses on the acceptance of membership in CBHIs rather than on a more general insurance system or on increasing resources through taxes. Finally, the study was conducted over the telephone and with a quantitative questionnaire, leaving little room for discussion. This reveals the need for subsequent qualitative studies to better understand, for example, the possible bias of social desirability, as mentioned in Burkina Faso [26]. This survey method over the telephone also reduced our ability to have more accurate household socio-economic data for further statistical analysis; the calculation of the standard of living based on the assets requested over the phone may explain the need for more relation to the variables of interest. Although our research is relatively descriptive and could not mobilize sophisticated statistical methods, the findings are beneficial for policymakers and researchers. For Senegalese policymakers, the findings respond to part of their demand for evidence to pursue current reforms in favour of greater insurance coverage. This evidence will inform the design of the new National Agency's strategic plan (2023–2027). For the scientific community, this study confirms the relevance of this topic and the need to finance and carry out research allowing for more sophisticated quantitative analyses and qualitative studies to understand better the challenges of the mandatory nature of insurance membership.

## Conclusion

As Senegal embarks on a critical UHC reform process for its risk pooling system, this research reveals the strong propensity of citizens to accept the mandatory nature of membership in mutual health insurance. If the declared rate is high, it poses significant implementation challenges and provides avenues for reflection for Senegalese policymakers in their quest for universal health coverage.

## Supporting information

**S1 Checklist. STROBE statement.**
(DOCX)

**S1 Questionnaire. Inclusivity in global research.**
(DOCX)

## Acknowledgments

We want to thank the respondents to the survey and the research assistants for their performance and dedication to the project: Clara Sadio, Khardiata Kane, Binetou Karfa Camara, Tabaski Diouf, and Coumba Sow. Thanks to Heather Hickey for the English proofreading.

## Author Contributions

**Conceptualization:** Valéry Ridde, Ibrahima Gaye, Bruno Ventelou, Elisabeth Paul, Adama Faye.

**Data curation:** Valéry Ridde, Ibrahima Gaye, Adama Faye.

**Formal analysis:** Valéry Ridde, Ibrahima Gaye, Bruno Ventelou, Elisabeth Paul, Adama Faye.

**Funding acquisition:** Valéry Ridde, Adama Faye.

**Investigation:** Valéry Ridde, Ibrahima Gaye, Bruno Ventelou, Adama Faye.

**Methodology:** Valéry Ridde, Ibrahima Gaye, Bruno Ventelou, Elisabeth Paul, Adama Faye.

**Project administration:** Valéry Ridde, Ibrahima Gaye, Adama Faye.

**Supervision:** Valéry Ridde, Ibrahima Gaye, Bruno Ventelou, Adama Faye.

**Validation:** Valéry Ridde, Ibrahima Gaye, Bruno Ventelou, Elisabeth Paul, Adama Faye.

**Writing – original draft:** Valéry Ridde.

**Writing – review & editing:** Valéry Ridde, Ibrahima Gaye, Bruno Ventelou, Elisabeth Paul, Adama Faye.

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
