## [Decision Letter · Decision Letter 0]

19 May 2023

PGPH-D-23-00600

Mandatory membership of community-based mutual health insurance in Senegal: A national survey

Dear Dr. Ridde,

Thank you for submitting your manuscript to PLOS Global Public Health. After careful consideration, we feel that it has merit but does not fully meet PLOS Global Public Health’s publication criteria as it currently stands. Therefore, we invite you to submit a revised version of the manuscript that addresses the points raised during the review process.

We look forward to receiving your revised manuscript.

Kind regards,

Osondu Ogbuoji

Academic Editor

Journal Requirements:

1. Please include the following request in the decision letter, and ping me with follow-up. “Please include a complete copy of PLOS’ questionnaire on inclusivity in global research in your revised manuscript. Our policy for research in this area aims to improve transparency in the reporting of research performed outside of researchers’ own country or community. The policy applies to researchers who have travelled to a different country to conduct research, research with Indigenous populations or their lands, and research on cultural artefacts. The questionnaire can also be requested at the journal’s discretion for any other submissions, even if these conditions are not met.  Please find more information on the policy and a link to download a blank copy of the questionnaire here: https://journals.plos.org/globalpublichealth/s/best-practices-in-research-reporting. Please upload a completed version of your questionnaire as Supporting Information when you resubmit your manuscript.”

2. Please send a completed 'Competing Interests' statement, including any COIs declared by your co-authors. If you have no competing interests to declare, please state "The authors have declared that no competing interests exist". Otherwise please declare all competing interests beginning with twhe statement "I have read the journal's policy and the authors of this manuscript have the following competing interests:"

Additional Editor Comments (if provided):

This is an important paper because of the potential contributions to UHC in Africa. It will need major revision before it can be accepted. Please submit a revision along with a completed STROBE checklist (Cross-sectional version found here: https://www.equator-network.org/reporting-guidelines/strobe/). Also address the comments from the reviewers below. Your paper will also benefit from an english language editor.

Reviewers' comments:

Reviewer's Responses to Questions

**Comments to the Author**

1. Does this manuscript meet PLOS Global Public Health’s publication criteria? Is the manuscript technically sound, and do the data support the conclusions? The manuscript must describe methodologically and ethically rigorous research with conclusions that are appropriately drawn based on the data presented.

Reviewer #1: Partly

Reviewer #2: Partly

2. Has the statistical analysis been performed appropriately and rigorously?

Reviewer #1: I don't know

Reviewer #2: No

3. Have the authors made all data underlying the findings in their manuscript fully available (please refer to the Data Availability Statement at the start of the manuscript PDF file)?

Reviewer #1: Yes

Reviewer #2: No

4. Is the manuscript presented in an intelligible fashion and written in standard English?

Reviewer #1: Yes

Reviewer #2: No

5. Review Comments to the Author

Reviewer #1: This is a very important study as African countries seek ways to promote UHC. However, there are a few observations and feedback you may consider in revising the manuscript.

In the introductory section, it may be useful to include the population of Senegal. It may also be useful to give information on the proportion of the population who own a mobile phone. This additional information will provide more context.

In line 126, the use of the words 'qualitative variable' may be misleading, and I suggest the use of 'categorical variable' as a replacement.

In the discussion section starting at line 180, I will suggest summarizing your results, before delving into comparisons with other studies.

In line 219 - 221, it is not quite clear to me what the state subsidies comprise of.

Is it possible to have respondents who did not speak any of the major languages? is it possible to explain what was planned and what was done?

It is also possible that some meanings may have been lost in the process of translation.

concerning the figures, it may be necessary to provide a legend and appropriate axis labels. the figure and tables may also be better presented with brief explanatory footnotes.

Have consideration been given to the use of regression models for data analysis? This may help identify associations of some predictor variable with measured outcomes and possible confounding.

Reviewer #2: The paper does not elaborate on objectives and specific hypotheses for testing through analysis of the quantitative survey dataset. There is no information on what is the current levels of health insurance coverage in the country for both formal and informal workers/households. Statistical analysis is very rudimentary. Hwoo the well-being is measured iis not explained. On line 87-88 the average age is 19 years (of whom, General population, participants) which is not correct. Discussion need better articulation.

6. PLOS authors have the option to publish the peer review history of their article (what does this mean?). If published, this will include your full peer review and any attached files.

**Do you want your identity to be public for this peer review?** For information about this choice, including consent withdrawal, please see our Privacy Policy.

Reviewer #1: No

Reviewer #2: **Yes: **Anil Gumber

---

## [Decision Letter · Decision Letter 1]

31 Aug 2023

Mandatory membership of community-based mutual health insurance in Senegal: A national survey

PGPH-D-23-00600R1

Dear Dr. Ridde,

We are pleased to inform you that your manuscript 'Mandatory membership of community-based mutual health insurance in Senegal: A national survey' has been provisionally accepted for publication in PLOS Global Public Health.

Best regards,

Julia Robinson

Executive Editor

Reviewer Comments (if any, and for reference):

Reviewer's Responses to Questions

**Comments to the Author**

1. If the authors have adequately addressed your comments raised in a previous round of review and you feel that this manuscript is now acceptable for publication, you may indicate that here to bypass the “Comments to the Author” section, enter your conflict of interest statement in the “Confidential to Editor” section, and submit your "Accept" recommendation.

Reviewer #2: All comments have been addressed

2. Does this manuscript meet PLOS Global Public Health’s publication criteria? Is the manuscript technically sound, and do the data support the conclusions? The manuscript must describe methodologically and ethically rigorous research with conclusions that are appropriately drawn based on the data presented.

Reviewer #2: Yes

3. Has the statistical analysis been performed appropriately and rigorously?

Reviewer #2: No

4. Have the authors made all data underlying the findings in their manuscript fully available (please refer to the Data Availability Statement at the start of the manuscript PDF file)?

Reviewer #2: Yes

5. Is the manuscript presented in an intelligible fashion and written in standard English?

Reviewer #2: Yes

6. Review Comments to the Author

Reviewer #2: Thanks for addressing and incorporating comments and suggestions

7. PLOS authors have the option to publish the peer review history of their article (what does this mean?). If published, this will include your full peer review and any attached files.

**Do you want your identity to be public for this peer review?** For information about this choice, including consent withdrawal, please see our Privacy Policy.

Reviewer #2: **Yes: **Anil Gumber
